# Gender Differences in the Factors Associated with Alcohol Binge Drinking: A Population-Based Analysis in a Latin American Country

**DOI:** 10.3390/ijerph19094931

**Published:** 2022-04-19

**Authors:** Akram Hernández-Vásquez, Horacio Chacón-Torrico, Rodrigo Vargas-Fernández, Leandro Nicolás Grendas, Guido Bendezu-Quispe

**Affiliations:** 1Centro de Excelencia en Investigaciones Económicas y Sociales en Salud, Vicerrectorado de Investigación, Universidad San Ignacio de Loyola, Lima 15024, Peru; 2Facultad de Ciencias de la Salud, Universidad Científica del Sur, Lima 15067, Peru; hchaconto@cientifica.edu.pe (H.C.-T.); jvargasf@ucientifica.edu.pe (R.V.-F.); 3Instituto de Farmacología, Facultad de Medicina, Universidad de Buenos Aires, Buenos Aires 1121, Argentina; leandrogrendas@hotmail.com; 4Centro de Investigación Epidemiológica en Salud Global, Universidad Privada Norbert Wiener, Lima 15046, Peru; guido.bendezu@uwiener.edu.pe

**Keywords:** alcohol drinking, alcoholism, binge drinking, health surveys, Peru

## Abstract

Alcohol consumption is a public health problem in Peru, fostered by traditional practices, where promoting social interaction in celebrations, facilitating field work as a source of energy and warmth, and achieving objectives in certain labor negotiations, play an important role. However, research on the risk factors of binge drinking according to gender is limited. The study aim was to determine the factors associated with binge drinking in the Peruvian adult population by gender. An analytical study of secondary data from the 2018 Peruvian Demographic and Family Health Survey was conducted. The dependent variable was binge drinking in the last 30 days. Adjusted prevalence ratios (aPR) were estimated for the association between sociodemographic and health-related variables with binge drinking. A total of 32,020 adults were included. Binge drinking was found in 22.4%. Men (32.6%; 95% confidence interval [CI]: 31.4–33.8) presented a higher consumption pattern compared to women (12.8%; 95% CI: 12.0–13.6). For both genders, differences were found in binge drinking according to sociodemographic characteristics (age and wealth quintile was associated in both genders while the educational level was associated only for men, and ethnic self-identification and marital status for women) and health- characteristics related (health insurance, smoking in the last 30 days, overweight and obesity were associated in both genders). Several factors are associated with binge drinking according to gender in the Peruvian population, including age and education level among men, as well as marital status and ethnic self-identification among women.

## 1. Introduction

Alcohol consumption is a significant public health problem globally. The World Health Organization reported that in 2016, 7.2% of premature deaths in people aged 69 and younger were related to alcohol consumption [1]. In the Americas, alcohol consumption has a higher than 50% lifetime prevalence, and it has been estimated that 5.5% of deaths and 6.7% of disability-adjusted life years (DALYs) were attributable to its consumption in 2016 [1]. Moreover, it has been described that, in addition to a high prevalence throughout life in this region, alcohol intake characteristically presents with irregular and excessive consumption patterns [2]. Binge drinking is especially harmful, taking into account that this consumption pattern increases the probability of developing more than 60 diseases and is responsible for more than three million deaths annually [3]. It is associated with violence, injuries, psychiatric disorders, suicidal behavior, poor academic performance, tobacco consumption, risky sexual behaviors, and alcohol-related driving offenses [1,4,5,6].

After the European region, the Americas region has the highest proportion of alcohol consumers, being the region with the lowest percentage of abstainers [1]. In the Americas, one out of every five current alcohol consumers has also had a binge-drinking episode at least once a month (above the world average of 16%) [7]. In particular, in the Latin American countries, there are only a few studies on this alcohol consumption pattern, given that it has been described as a risk factor for depression or cardiovascular disease [8,9]. It is also interesting to note that part of the Latin American countries consumes considerable amounts of low-quality alcohol, which can have harmful consequences for consumer health [10,11]. In this region, the proportion of men with binge drinking consumption is greater than in women (two times greater in most countries) [1].

In Peru, one of the countries in this region, the per capita consumption of pure alcohol has been estimated as 6.3 L per year, being close to the global average and lower than the 8 L regional average [1]. However, in 2016, alcohol dependence was a public health problem leading the DALYs by disease burden in Peru and ranking sixth (approximately 5 DALYs per 1000 inhabitants) in the population aged 15 to 44 years [12]. In this country, the consumption of traditional alcoholic beverages at an early age is socially accepted, promoting alcohol drinking patterns [13].

Literature shows that there are gender differences in binge drinking. It has been reported that men are nearly twice as likely to follow this consumption pattern compared to women [14]. Consequently, research on binge drinking in women has been underrepresented, however, when they develop this disorder, they tend to exhibit more severe symptoms than men [15]. Indeed, a prior meta-analysis showed that females had a higher risk of mortality compared to men among heavy drinkers [16]. An analysis of a Korean National Survey also showed that young age, lower educational level, and unmarried status are associated factors with heavy alcohol consumption [17]. A recent systematic review found that depression, partner’s alcohol use, and unplanned pregnancy were associated factors with alcohol consumption during pregnancy [18]. The gender-based consumption gap is conditioned by psychological, social, and cultural factors, which vary over time and according to diverse biological factors [19,20]. Cultural differences have been shown to influence the pattern of alcohol consumption from young to old; additionally, binge drinking is seen to have a different effect according to ethnic group differences [20,21,22]. Likewise, marital status, age, religious identity, and socioeconomic status have also been associated with binge drinking [19].

Diverging trends in alcohol drinking patterns have been observed around the world in recent years and, in turn, have increased substantially in several lower-middle-income countries [23]. Few population-based studies have compared associated factors of binge drinking between women and men in the same analysis. Therefore, comparing results from different papers is not reliable due to heterogeneous contexts (i.e., inclusion criteria). Moreover, by assessing the binge drinking consumption associated factors between genders, we could identify risk clusters. This information will help promote better health policies to reduce consumption incidence in those groups. Therefore, this study aimed to determine the factors associated with binge drinking in the Peruvian adult population according to gender. We hypothesized that men would have larger binge drinking consumption patterns than women and that the association would substantially vary according to gender.

## 2. Materials and Methods

### 2.1. Design and Study Population

An analytical study of secondary data from the 2018 Demographic and Family Health Survey (ENDES, by its acronym in Spanish) was conducted. ENDES is a Peruvian population-based survey that collects information on lifestyle, maternal and child health, non-communicable diseases, and health services-related variables [24]. Since 1986, the National Institute of Statistics and Informatics (INEI, by its acronym in Spanish) has been carrying out ENDES under the MEASURE-DHS model, promoted by the Demographic and Health Surveys Program (DHS) [25].

ENDES uses balanced, two-stage, stratified and independent probability sampling at a departmental level and by urban and rural areas to obtain representative annual estimates at the national, urban/rural level by geographic domain (Metropolitan Lima, Coast, Andean, and Amazon) and by administrative regions (25 regions). Thus, the results obtained from the analysis of the annual ENDES data provide representative estimates at the following levels: national (total Peruvian population), urban and rural, natural regions, and administrative regions [24]. The ENDES is made up of three questionnaires: (1) a household questionnaire (collects information on the household and its members); (2) an individual women’s questionnaire (collects information on reproductive health and maternal and child health); and (3) a health questionnaire (aimed at people aged 15 years and over and collects information on chronic non-communicable diseases and their risk factors, among which is the pattern of alcohol consumption). The processing of ENDES data to study binge drinking in the Peruvian adult population was carried out following what was described in a previous study [26]. More information on the ENDES survey can be found on the INEI website [24].

Regarding the sample size, a total of 32,020 people were included in the analysis. Among the inclusion criteria, people aged 18 years or older, with complete information on alcohol consumption in the last 30 days, and complete data on the health questionnaire were considered. Among the exclusion criteria, the reasons for which the participants could not carry out the survey were considered (absence, rejection of the interview, incomplete data, disability, among others). Finally, a total of 6712 people with binge drinking were determined, which is equivalent to 22.4% of the entire population included in the analysis (Figure 1).

### 2.2. Variables and Measurements

The dependent variable was binge drinking, which was defined by the United States Substance Abuse and Mental Health Services Administration (SAMHSA) as the consumption of five and four or more alcoholic beverages on the same occasion for men and women, respectively, in the last 30 days prior to the survey [27]. For the estimation of prevalence rates, the dependent variable was dichotomized into binge drinking (coded as 1) and non-consumption (coded as 0) regarding the last 30 days.

The following independent variables were considered, chosen, and constructed based on epidemiological criteria as reported in similar studies on binge drinking and alcohol consumption and in studies based on the DHS [28,29,30,31,32,33]. To describe the sociodemographic characteristics of the population, the main independent variable was gender (male/female). Other characteristics included were age (18–24/25–44/45–59/60 or more years); natural regions (Coast/Andean/Amazon); ethnicity (non-native/native/Afro-Peruvian); marital status (single/married or cohabiting/separated, divorced or widowed); education (without education/primary/secondary/higher); wealth quintile (quintile 1 (poorest)/quintile 2/quintile 3/quintile 4/quintile 5 (richest)); and area of residence (urban/rural). In the ethnicity variable, the native category is made up of those people who identify themselves as belonging to an indigenous ethnic group in Peru, the non-native category is made up of people who identify themselves as white, mestizo, or other, while the category Afro-Peruvian includes people self-identified as black, brown, zambo, mulatto, or Afro-Peruvian people. To describe health-related characteristics of the population included in the study, the variables used were smoker (a person who consumed cigarettes in the 30 days prior to the survey was considered to be a smoker and the variable was dichotomized in Yes (when cigarettes were consumed) and No (when cigarettes were not consumed)); consumption of fruits and vegetables (Yes/No (five or more fruits and vegetables per day)); depressive symptomatology (defined as the presence of depressive symptoms in the 14 days prior to the survey (determined by a score of five or more in the Patient Health Questionnaire-9 (PHQ-9) screening test) [34], being dichotomized in Yes (when fulfilling the operational definition) and No (when no depressive symptoms were presented, and the PHQ-9 test score was less than five)); body mass index (BMI, with categories: thinness to normal/overweight/obesity); high blood pressure (Yes/No), history of diabetes (Yes/No); and health insurance (Yes/No). Regarding health insurance, it was considered yes if the individual was affiliated with Seguro Integral de Salud (for the poor and extremely poor, approximately 45% of the population), or Social Health Insurance (for dependent workers and their legal beneficiaries, approx. 25% of the insured population), or Police, Armed forces, or private health insurance providers (approx. 5% of the insured population) [35]. In other cases, it was considered to be ‘no health insurance’.

### 2.3. Statistical Analysis

All variables of interest for the study were stratified according to gender. For the univariate analysis, the absolute frequencies and weighted proportions of independent variables were reported to characterize the population. In the bivariate analysis, the association between independent variables and binge drinking was evaluated using the Chi-squared test. Finally, to determine the sociodemographic and health-related factors associated with binge drinking in Peruvian adults, the crude (PR) and adjusted (aPR) prevalence ratios were estimated with their respective 95% confidence intervals (95% CI) as measures of association. It should be noted that the fitted model was performed considering independent variables that resulted in a *p*-value < 0.05 in the crude analysis. Poisson’s family generalized linear models (GLM) with link function (log) were used for the crude and adjusted analyses, and a value of *p* < 0.05 was considered statistically significant. The analyses were carried out using the Stata 14.2 statistical program (Stata Corp, College Station, TX, USA).

### 2.4. Ethical Considerations

This study did not require the approval of an ethics committee as it was a secondary data analysis from a freely accessible and public domain database, which does not allow the identification of the evaluated participants. The ENDES 2018 databases can be accessed and downloaded from the INEI web portal (http://iinei.inei.gob.pe/microdatos, accessed on 10 November 2021) or through the *ENDES.PE* R package [26].

## 3. Results

### 3.1. General Characteristics of the Study Sample

The total number of Peruvian adult individuals included in the analysis was 32,020, of whom 18,308 (57.2%) were women. The predominant age group was 25–44 years, with 42.8% and 42.4% men and women, respectively. The background characteristics of respondents by gender are presented in Table 1.

### 3.2. Binge Drinking Prevalence

Binge drinking was reported in 22.4% (95% CI: 21.7–23.2) of the general adult population, being 12.8% (95% CI: 12.0–13.6) in women and 32.6% (95% CI: 31.4–33.8) in men. Table 2 shows the weighted proportions of binge drinking in men and women according to sociodemographic characteristics and related to health status. In general, men had higher proportions of binge drinking for all independent variables and categories included in the study. The youngest groups presented the highest prevalence of binge drinking in both men (18–24 years: 30.3%, 95% CI: 27.3–33.5; 25-44 years: 40.7%, 95% CI: 39.0–42.4) and women (18–24 years: 15.6, 95% CI: 13.5–17.9; 25–44 years: 15.7%, 95% CI: 14.5–16.8). Regarding education and wealth, a higher educational level or wealth quintile was associated with a higher rate of binge drinking in both genders. Regarding health-related variables, a lack of health insurance, having a BMI classified as overweight or obese, or smoking habit presented higher proportions of binge drinking in both men and women. In addition, binge drinking was more frequent in women with hypertension.

### 3.3. Sociodemographic Characteristics and Binge Drinking

Regarding the sociodemographic variables associated with binge drinking, men aged 25–44 had a higher probability of binge drinking (aPR: 1.28; 95% CI: 1.14–1.45), while the age group of 60 years or older was associated with a lower probability (aPR: 0.70; 95% CI: 0.58–0.84) of binge drinking compared to the younger group of men (18 to 24 years). In women, the age group of 60 years or older was associated with a lower probability of this consumption pattern (aPR: 0.24; 95% CI: 0.17–0.34). Education was a factor associated with a higher prevalence of binge drinking in men, especially in those with a secondary (aPR: 2.01; 95% CI: 1.34–3.02) or higher level of education (aPR: 2.04; 95% CI: 1.35–3.08) while this characteristic was not found to be associated with binge drinking in women. Belonging to the wealth quintiles 2, 4 and 5 in men was found to be associated with a higher probability of binge drinking compared to belonging to the poorest quintile (quintile 1). In women, wealth quintiles 2 to 5 were found to be associated with a higher probability of binge drinking compared to belonging to the poorest quintile (quintile 1). Although in men ethnic self-identification was not associated with binge drinking, Afro-Peruvian ethnicity (aPR: 1.26; 95% CI: 1.04–1.53) was associated with a higher probability of binge drinking in females. Furthermore, while marital status was not found to be associated in men with binge drinking, married/cohabiting status (aPR: 0.81; 95% CI: 0.69–0.96) was associated with lower binge drinking in women (Table 3). On characteristics such as the area of residence and natural region, these were not associated with binge drinking in men or women.

### 3.4. Health-Related Characteristics and Binge Drinking

Regarding health-related characteristics, not having health insurance increased the probability of binge drinking in both men (aPR: 1.12; 95% CI: 1.04–1.21) and women (aPR: 1.14; 95% CI: 1.00–1.30). Males (aPR: 1.96; 95% CI: 1.83–2.10) and females that had smoked cigarettes in the last 30 days (aPR: 2.87; 95% CI: 2.46–3.35) also had a higher probability of binge drinking. Overweight and obesity were found to be associated with binge drinking in men (aPR: 1.12; 95% CI: 1.03–1.22 and aPR: 1.22; 95% CI: 1.11–1.34, respectively) and women (aPR: 1.30; 95% CI: 1.12–1.50 and aPR: 1.50; 95% CI: 1.28–1.76, respectively). The consumption pattern of five or more fruits and vegetables per day, the presence of depressive symptoms, or having hypertension or the diabetes history were not associated with binge drinking in either men or women (Table 3).

## 4. Discussion

This study sought to determine the factors associated with binge drinking in the Peruvian adult population according to gender. Binge drinking was detected in one-fifth of the population. With respect to our hypothesis, it was shown that men consume more than women since the binge drinking pattern was higher in males (three out of ten).

We found that men had a higher proportion of binge drinking compared to women. A previous study carried out in one of the regions with the largest Peruvian population, described a similar prevalence of binge drinking [8]. Prior regional and multinational estimates of heavy drinking evaluating gender differences found consistently higher proportions among men than women [36,37]. We again confirmed this by describing how binge drinking consumption patterns are considerably higher in men than in women among our Peruvian population. Apparently, biological as well as cultural differences are among the main drivers of these differences that need to be further studied [36]. Considering that alcohol is one of the most harmful drugs for health [38,39,40], the relevance of this study consists of providing evidence regarding binge drinking in a Latin American population, because binge drinking pattern behaviors represent a tremendous detrimental burden to health [41].

Likewise, it was identified that the 60 years or older individuals had a lower probability of binge drinking than the 18 to 24-year-old subjects in both genders, whereas the decline in the likelihood of binge drinking was more significant among women than in men. In Peru, alcohol consumption usually begins in adolescence, with a higher proportion of men initiating this consumption according to a historical pattern of higher acceptance among this population [42]. Our findings are in line with what is reported in the literature, which indicates that adolescents and young adults are the population group at the highest risk of this disorder [6,29]. Indeed, a study in nine countries of different income and cultural characteristics found that younger age groups were associated with a greater likelihood of high-risk drinking [37]. Furthermore, it has been reported that being 50 years or older was associated with a lower prevalence of binge drinking patterns, such as heavy drinking [43]. Thus, the implementation of strategies aimed at controlling binge drinking in adolescents and young adults is required. For instance, the National Drug Control Strategy 2017–2021 was established in Peru to promote healthy lifestyles, strengthen policies regulating alcohol consumption, and develop preventive programs for specific populations [12].

According to socioeconomic status (SES), the population with the highest wealth quintile (quintile 5) had the highest probability of presenting this disorder, with the association being higher in women than in men. Furthermore, there is a socioeconomic gradient strongly associated with alcohol consumption, in which the population within a lower wealth quintile (quintile 1) presents a lower proportion of alcohol consumption and vice-versa [21,44,45]. Additionally, a study in Singapore found that greater annual household income was associated with higher lifetime and twelve-month heavy drinking [46]. Although it is described that in Latin America and the Caribbean, the price of alcohol is low and affordable for the major proportion of the population [11], the found differences in the binge drinking pattern among wealthy and poorest groups could be explained since most people in low- and middle-income countries simply do not possess the purchasing power to buy alcohol for heavy episodic drinking [47]. On the other hand, people with higher income may socialize more often, making this pattern of alcohol consumption within social norms more likely [48]. The literature reports the alcohol harm paradox, which notes that the populations within the lowest socioeconomic strata are those with the highest mortality and morbidity related to alcohol, despite having the lowest consumption [49,50]. The lack of protection against alcohol damage due to consumption in unsafe spaces, lower access to health care services, and weak social support networks could explain the higher alcohol-related health burden in the population with fewer resources [1]. Another study finding was that men and women without health insurance have a greater prevalence of binge drinking. In Peru, approximately three-quarters of the population has some type of health insurance and the insured population, are mostly young adults [35]. This finding is confirmed by what was reported in our study on the age of the participants, where the young adult population presents a higher probability of binge drinking as opposed to older adults. This indicates that a subgroup of the population presents a barrier to access health care and has limited access to health promotion and addiction prevention activities necessary in people suffering from binge drinking.

Concerning education, men with primary or higher education levels were more likely to binge drink than those without formal education. In the international literature, adults with secondary or higher education levels have shown a higher prevalence of binge drinking [51,52], problem drinking [53], and the demand for alcohol increases with increasing education [54]. It is imperative to mention that this relationship was not evident in women in the present study. Explanations in this regard have been hypothesized, including the perspective and gender role of women in society and social gradients that differ according to different generational cohorts [44]. The absence of influence of the education level in binge drinking patterns in women found in the present study could be explained by social and cultural gender role differences in the Peruvian population, in which alcohol consumption by males is accepted and promoted while consumption by females is criticized and considered as inappropriate or an uneducated behavior [42]. Regardless of the mechanism underlying this finding, it is necessary to study these gender differences more in-depth to develop binge drinking prevention programs addressing populations of various educational backgrounds, with particular emphasis on binge drinking vulnerability among men with higher educational backgrounds.

Afro-Peruvian ethnic self-identification increased the probability of binge drinking in women, but not in men. The literature in this regard is variable since studies in the US population found that non-Hispanic African Americans [55] and specifically African American women [56] had one of the lowest proportions of binge drinking. On the other hand, in Brazil, there has been reported no difference in harmful alcohol consumption patterns in women according to ethnic identification [21]. Since ethnicity could be a contributing factor in alcohol binge drinking patterns depending on the type of society, it would be necessary to expand the research of the factors associated with binge drinking in this population, including the influence of this factor. Additionally, married or cohabiting women had a lower probability of binge drinking. Previous studies have evidenced that married women have been reported to be less likely to binge drink than unmarried women [17,57]. This finding could be explained in that compared to being single, marriage or cohabitation is related to a lower occurrence of stress, depression, loneliness, and alcoholism, suggesting better mental health characteristics in this population [58].

As for health-related variables, smokers were found to have a nearly two (male) and three (women) times higher probability of binge drinking than the non-smoking population. The concurrent use of both addictive substances is widely described in the literature, with smoking being associated with binge drinking in university students from different regions of the world [59,60]. In specific contexts, such as social events, simultaneous consumption of alcohol and tobacco occurs since they share explanatory mechanisms of their use in these activities [61]. Thus, both social behaviors in the Peruvian adult population are below what is described globally. On the other hand, overweight or obese men and women were more likely to present a binge drinking pattern. It has been reported that binge drinking in episodes increases the risk of becoming overweight and developing obesity [62], which may explain the said finding.

Regarding the limitations of the study, being a study of secondary data, there is the possibility of a lack of accurate data. Besides, since the survey collects data on alcohol consumption in the last 30 days, and since ENDES is a self-reporting survey, respondents may not have accurately remembered their consumption of alcoholic beverages, and thus, the results could be affected by recall bias. Moreover, as it is a harmful habit for the individual and society, the results could be affected by a social desirability bias. Similarly, some variables of interest for the study of alcohol consumption patterns in the population, including socioeconomic and psychosocial factors such as labor force enrollment, patterns of alcohol consumption at home, positive expectations towards alcohol consumption, among others, were not measured or the information available in ENDES database was incomplete or not suitable to be included in the analysis, thereby limiting better characterization of the phenomenon studied. Nonetheless, we consider that the study findings are useful as an approximation to the study of binge drinking in the Peruvian adult population, given that ENDES is a nationally representative survey that has quality control processes and is widely used for the study of health topics in the Peruvian population. We have compared our data with studies from other parts of the world and found comparable results. Furthermore, the study results could be relevant for other Latin American countries since there are common alcohol consumption patterns in this region.

## 5. Conclusions

Binge drinking occurs in one-fifth of Peruvian adults with a higher proportion in men (three out of ten compared to one out of ten in women), with sociodemographic and health-related variables being associated with this consumption pattern according to gender. Differences were found in binge drinking according to sociodemographic characteristics (age and wealth quintile in both genders, and separately the educational level in men and ethnic self-identification and marital status in women) and health-related characteristics (health insurance, smoking in the last 30 days, being overweight and obesity in both genders) in men and women. The consequences of the harmful use of alcohol can be reduced through interventions aimed directly at demographic and socioeconomic contexts showing differential vulnerability and exposure. To our knowledge, this is the first study that reports the differential impact of factors associated with binge drinking according to gender at a national level in a Latin American population. Interventions against this consumption pattern require new and novel approaches that take into account the specific factors of each region. Given the association of this consumption pattern with interpersonal violence, traffic accidents, and chronic health problems due to alcohol consumption and the increase in health costs, studies are needed to evaluate how and to what extent binge drinking affects the population and the health systems, and strategies should also be developed to mitigate binge drinking in groups of higher vulnerability in the Latin America region.

## Figures and Tables

**Figure 1 ijerph-19-04931-f001:**
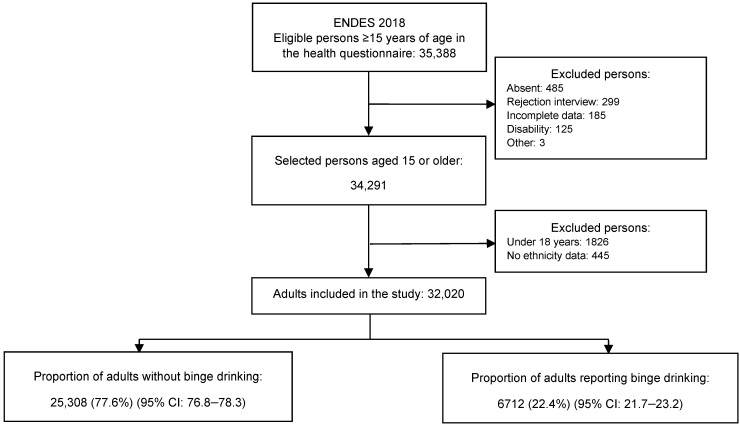
Flowchart of the selection of adults included in the study.

**Table 1 ijerph-19-04931-t001:** Background characteristics of respondents by gender (*n* = 32,020).

Characteristics	Total (*n* = 32,020)		Men		Women	
	*n*	Weighted Proportion *	*n*	Weighted Proportion *	*n*	Weighted Proportion *
Sample size:	32,020	100	13,712	100	18,308	100
Age groups, years:						
18–24	4613	16.5 (15.8–17.2)	1729	16.8 (15.7–17.8)	2884	16.2 (15.3–17.0)
25–44	16,776	42.6 (41.7–43.5)	7063	42.8 (41.5–44.1)	9713	42.4 (41.4–43.5)
45–59	5757	22.6 (21.8–23.4)	2717	22.6 (21.4–23.7)	3040	22.6 (21.5–23.7)
60 or more	4874	18.4 (17.7–19.1)	2203	17.9 (16.9–19.0)	2671	18.8 (17.9–19.8)
Natural regions:						
Amazon	7321	11.9 (11.3–12.5)	3210	12.3 (11.6–13.1)	4111	11.5 (10.9–12.2)
Andean	11,660	24.8 (23.9–25.8)	4874	24.3 (23.2–25.5)	6786	25.2 (24.1–26.3)
Coast	13,039	63.3 (62.4–64.2)	5628	63.3 (62.1–64.5)	7411	63.3 (62.2–64.3)
Area:						
Urban	21,043	80.8 (80.3–81.3)	8833	80.4 (79.6–81.2)	12,210	81.1 (80.5–81.8)
Rural	10,977	19.2 (18.7–19.7)	4879	19.6 (18.8–20.4)	6098	18.9 (18.2–19.5)
Ethnicity:						
Non-native	17,477	64.0 (63.1–64.8)	7468	63.4 (62.2–64.6)	10,009	64.6 (63.5–65.5)
Native	11,742	26.7 (26.0–27.5)	4993	26.8 (25.7–28.0)	6749	27.7 (25.8–27.7)
Afro-Peruvian	2801	9.3 (8.7–9.8)	1251	9.8 (9.1–10.5)	1550	8.8 (8.1–9.4)
Marital status:						
Never married	4345	17.3 (16.6–18.1)	2197	19.9 (18.8–21.0)	2148	14.9 (14.0–15.8)
Married/Cohabiting	22,597	66.0 (65.1–66.8)	10,214	69.6 (68.3–70.8)	12,383	62.6 (61.4–63.7)
Separated/Divorced/Widowed	5078	16.7 (16.1–17.4)	1301	10.5 (9.7–11.4)	3777	22.6 (21.5–23.6)
Education level:						
No formal school	1717	4.2 (3.9–4.5)	278	1.6 (1.4–2.0)	1439	6.6 (6.1–7.1)
Primary	8036	20.6 (10.0–21.3)	3243	18.2 (17.3–19.1)	4793	22.9 (22.1–23.8)
Secondary	12,842	40.6 (39.7–41.4)	6009	44.0 (42.6–45.3)	6833	37.4 (36.3–38.5)
Higher	9425	34.6 (33.7–35.5)	4182	36.2 (34.8–37.5)	5243	33.1 (32.0–34.2)
Wealth Index:						
Poorest	10,121	18.2 (17.7–18.8)	4412	18.3 (17.5–19.1)	5709	18.2 (17.5–18.9)
Poorer	7903	20.4 (19.7–21.2)	3388	21.1 (20.0–22.2)	4515	19.8 (18.9–20.7)
Middle	5938	20.8 (20.1–21.6)	2469	20.3 (19.3–21.4)	3469	21.3 (20.3–22.4)
Richer	4615	20.6 (19.8–21.4)	1992	21.0 (19.9–22.3)	2623	20.2 (19.2–21.2)
Richest	3443	19.9 (19.0–20.8)	1451	19.3 (18.0–20.6)	1992	20.5 (19.5–21.6)
Health insurance:						
Yes	24,367	72.7 (71.8–73.5)	9675	68.2 (67.0–69.5)	14,692	76.9 (75.8–77.9)
No	7653	27.3 (26.5–28.2)	4037	31.8 (30.5–33.0)	3616	23.1 (22.1–24.2)
Smoked cigarettes in the last 30 days:						
No	28,724	88.3 (87.6–88.9)	10,995	80.9 (79.8–82.0)	17,729	95.2 (94.6–95.7)
Yes	3296	11.7 (11.1–12.4)	2717	19.1 (18.0–20.2)	579	4.8 (4.3–5.4)
Fruits and vegetables 5 or more:						
No	29,036	88.9 (88.3–89.5)	12,573	89.9 (89.1–90.7)	16,463	87.9 (87.1–88.7)
Yes	2984	11.1 (10.5–11.7)	1139	10.1 (9.3–10.9)	1845	12.1 (11.3–12.9)
PHQ-9 5 or more:						
No	25,150	79.6 (78.9–80.3)	11,599	85.6 (84.7–86.5)	13,551	73.9 (72.9–75.0)
Yes	6870	20.4 (19.7–21.1)	2113	14.4 (13.5–15.3)	4757	26.1 (25.0–27.1)
Body mass index:						
Thinness to Normal	12,157	36.1 (35.3–37.0)	5902	39.1 (37.8–40.4)	6255	33.4 (32.3–34.5)
Overweight	12,586	39.3 (38.4–40.2)	5366	39.8 (38.5–41.1)	7220	38.8 (37.7–39.9)
Obesity	7277	24.6 (23.8–25.4)	2444	21.1 (20.1–22.2)	4833	27.8 (26.8–28.9)
Hypertension: **						
No	26,310	78.7 (78.0–79.5)	10,805	76.5 (75.3–77.6)	15,505	80.9 (79.9–81.8)
Yes	5710	21.3 (20.5–22.0)	2907	23.5 (22.4–24.7)	2803	19.1 (18.2–20.1)
Diabetes history:						
No	31,097	96.2 (95.8–96.5)	13,333	96.5 (95.9–97.0)	17,769	95.9 (95.4–96.4)
Yes	923	3.8 (3.5–4.2)	379	3.5 (3.0–4.1)	544	4.1 (3.6–4.6)

* Estimates included the expansion factor and ENDES sample specifications. ** If the average systolic blood pressure (two readings) was ≥140 mmHg or diastolic blood pressure was ≥90 mmHg or if hypertension had been previously diagnosed by a doctor.

**Table 2 ijerph-19-04931-t002:** Proportion of binge drinking among men and women according to sociodemographic and health-related characteristics.

Characteristics	Men		*p* Value	Women		
	No (Weighted Proportion *)	Yes (Weighted Proportion *)		No (Weighted Proportion *)	Yes (Weighted Proportion *)	*p* Value **
Overall	67.4 (66.2–68.6)	32.6 (31.4–33.8)		87.2 (86.4–88.0)	12.8 (12.0–13.6)	
Age groups, years:						
18–24	69.7 (66.5–72.7)	30.3 (27.3–33.5)	<0.001	84.4 (82.1–86.5)	15.6 (13.5–17.9)	<0.001
25–44	59.3 (57.6–61.0)	40.7 (39.0–42.4)		84.3 (83.2–85.5)	15.7 (14.5–16.8)	
45–59	69.1 (66.5–71.7)	30.9 (28.3–33.5)		86.8 (84.7–88.6)	13.2 (11.4–15.3)	
60 or more	82.4 (79.9–84.6)	17.6 (15.4–20.1)		96.8 (95.6–97.6)	3.2 (2.4–4.4)	
Natural regions:						
Amazon	67.2 (65.0–69.3)	32.8 (30.7–35.0)	<0.001	90.4 (89.2–91.5)	9.6 (8.5–10.8)	<0.001
Andean	69.3 (67.4–71.2)	30.7 (28.8–32.6)		89.6 (88.5–90.7)	10.4 (9.3–11.5)	
Coast	66.7 (64.9–68.3)	33.3 (31.7–35.1)		85.7 (84.5–86.9)	14.3 (13.1–15.5)	
Area:						
Urban	66.0 (64.5–67.4)	34.0 (32.6–35.5)	<0.001	85.7 (84.7–86.6)	14.3 (13.4–15.3)	<0.001
Rural	73.3 (71.4–75.0)	26.7 (25.0–28.6)		93.9 (93.0–94.7)	6.1 (5.3–7.0)	
Ethnicity:						
Non-native	66.0 (64.3–67.6)	34.0 (32.4–35.7)	<0.001	86.7 (85.7–87.7)	13.3 (12.3–14.3)	0.021
Native	70.6 (68.5–72.5)	29.4 (27.5–31.5)		88.9 (87.6–90.1)	11.1 (9.9–12.4)	
Afro-Peruvian	67.6 (63.8–71.1)	32.4 (28.9–36.2)		86.1 (83.4–88.5)	13.9 (11.5–16.6)	
Marital status:						
Never married	66.8 (63.8–69.5)	33.2 (30.5–36.2)	0.002	82.2 (79.5–84.7)	17.8 (15.3–20.5)	<0.001
Married/Cohabiting	66.6 (65.2–67.9)	33.4 (32.1–34.8)		88.1 (87.1–89.0)	11.9 (11.0–12.9)	
Separated/Divorced/Widowed	73.9 (70.2–77.3)	26.1 (22.7–29.8)		88.3 (86.4–89.9)	11.7 (10.1–13.6)	
Education level:						
No formal school	90.0 (85.1–93.4)	10.0 (6.6–14.9)	<0.001	96.9 (95.4–97.9)	3.1 (2.1–4.6)	<0.001
Primary	77.1 (74.8–79.2)	22.9 (20.8–25.2)		92.8 (91.5–93.9)	7.2 (6.1–8.5)	
Secondary	66.0 (64.2–67.8)	34.0 (32.2–35.8)		85.8 (84.4–87.2)	14.2 (12.8–15.6)	
Higher	63.2 (61.0–65.3)	36.8 (34.7–39.0)		83.1 (81.5–84.6)	16.9 (15.4–18.5)	
Wealth Index:						
Poorest	75.2 (73.3–77.0)	24.8 (23.0–26.7)	<0.001	95.0 (94.1–95.7)	5.0 (4.3–5.9)	<0.001
Poorer	67.5 (65.0–69.8)	32.5 (30.2–35.0)		90.1 (88.8–91.3)	9.9 (8.7–11.2)	
Middle	68.3 (65.6–70.9)	31.7 (29.1–34.4)		87.3 (85.6–88.8)	12.7 (11.2–14.4)	
Richer	64.1 (61.2–66.8)	35.9 (33.2–38.8)		82.3 (80.1–84.3)	17.7 (15.7–19.9)	
Richest	62.5 (58.8–66.0)	37.5 (34.0–41.2)		82.5 (80.2–84.6)	17.5 (15.4–19.8)	
Health insurance:						
Yes	69.3 (67.8–70.7)	30.7 (29.3–32.2)	<0.001	88.3 (87.5–89.1)	11.7 (10.9–12.5)	<0.001
No	63.4 (61.2–65.5)	36.6 (34.5–38.8)		83.7 (81.7–85.5)	16.3 (14.5–18.3)	
Smoked cigarettes in the last 30 days:						
No	73.2 (71.9–74.4)	26.8 (25.6–28.1)	<0.001	88.8 (88.0–89.5)	11.2 (10.5–12.0)	<0.001
Yes	42.9 (40.0–45.9)	57.1 (54.1–60.0)		56.5 (50.4–62.4)	43.5 (37.6–49.6)	
Fruits and vegetables 5 or more:						
No	67.4 (66.1–68.6)	32.6 (31.4–33.9)	0.969	87.3 (86.4–88.1)	12.7 (11.9–13.6)	0.874
Yes	67.5 (63.4–71.3)	32.5 (28.7–36.6)		87.1 (84.7–89.1)	12.9 (10.9–15.3)	
PHQ-9 5 or more:						
No	67.1 (65.8–68.4)	32.9 (31.6–34.2)	0.332	87.3 (86.4–88.2)	12.7 (11.8–13.6)	0.687
Yes	68.8 (65.7–71.8)	31.2 (28.2–34.3)		87.0 (85.3–88.5)	13.0 (11.5–14.7)	
BMI:						
Thinness to Normal	73.3 (71.7–75.0)	26.7 (25.0–28.3)	<0.001	90.4 (89.3–91.5)	9.6 (8.5–10.7)	<0.001
Overweight	65.6 (63.6–67.5)	34.4 (32.5–36.4)		86.5 (85.2–87.8)	13.5 (12.2–14.8)	
Obesity	59.7 (56.8–62.5)	40.3 (37.5–43.2)		84.4 (82.7–86.0)	15.6 (14.0–17.3)	
Hypertension: ***						
No	67.0 (65.6–68.3)	33.0 (31.7–34.4)	0.235	86.1 (85.2–87.0)	13.9 (13.0–14.8)	<0.001
Yes	68.7 (66.2–71.1)	31.3 (28.9–33.8)		91.9 (90.3–93.3)	8.1 (6.7–9.7)	
Diabetes history:						
No	67.2 (65.9–68.4)	32.8 (31.6–34.1)	0.068	87.2 (86.3–87.9)	12.8 (12.1–13.7)	0.315
Yes	73.8 (66.7–79.9)	26.2 (20.1–33.3)		89.5 (84.7–92.9)	10.5 (7.1–15.3)	

* Estimates included the expansion factor and ENDES sample specifications. ** *p* value was obtained using the Chi-square test with Rao-Scott adjustment. *** If its average systolic blood pressure (two readings) was ≥140 mmHg or diastolic blood pressure was ≥90 mmHg or if it had a previous diagnosis by a doctor.

**Table 3 ijerph-19-04931-t003:** Crude and adjusted prevalence ratios among men and women for several socioeconomic and health-related variables.

Characteristics	Men				Women			
	Crude Model *		Adjusted Model *		Crude Model *		Adjusted Model *	
	PR (95% CI)	*p*-Value	aPR (95% CI)	*p*-Value	PR (95% CI)	*p*-Value	aPR (95% CI)	*p*-Value
Age groups, years:								
18–24	Ref.		Ref.		Ref.		Ref.	
25–44	1.34 (1.20–1.50)	<0.001	1.28 (1.14–1.45)	<0.001	1.01 (0.86–1.18)	0.936	1.03 (0.88–1.22)	0.699
45–59	1.02 (0.89–1.16)	0.797	1.03 (0.89–1.19)	0.686	0.85 (0.70–1.04)	0.105	0.84 (0.68–1.02)	0.085
60 or more	0.58 (0.49–0.69)	<0.001	0.70 (0.58–0.84)	<0.001	0.21 (0.15–0.29)	<0.001	0.24 (0.17–0.34)	<0.001
Natural regions:								
Amazon	Ref.		No include		Ref.		Ref.	
Andean	0.93 (0.85–1.02)	0.145			1.09 (0.92–1.28)	0.32	1.05 (0.89–1.23)	0.564
Coast	1.02 (0.94–1.10)	0.696			1.49 (1.29–1.72)	<0.001	0.88 (0.76–1.02)	0.09
Area:								
Urban	Ref.		Ref.		Ref.		Ref.	
Rural	0.79 (0.72–0.85)	<0.001	1.06 (0.95–1.17)	0.302	0.43 (0.37–0.50)	<0.001	0.95 (0.78–1.16)	0.613
Ethnicity:								
Non-native	Ref.		Ref.		Ref.		Ref.	
Native	0.87 (0.79–0.94)	0.001	0.96 (0.89–1.05)	0.392	0.83 (0.73–0.95)	0.008	1.08 (0.94–1.24)	0.26
Afro-Peruvian	0.95 (0.84–1.08)	0.455	1.03 (0.92–1.16)	0.563	1.04 (0.86–1.27)	0.675	1.26 (1.04–1.53)	0.018
Marital status:								
Never married	Ref.		Ref.		Ref.		Ref.	
Married/Cohabiting	1.01 (0.91–1.11)	0.911	1.05 (0.94–1.16)	0.399	0.67 (0.57–0.79)	<0.001	0.81 (0.69–0.96)	0.014
Separated/Divorced/Widowed	0.78 (0.67–0.92)	0.002	1.03 (0.88–1.21)	0.686	0.66 (0.54–0.81)	<0.001	1.03 (0.85–1.25)	0.77
Education level:								
No formal school	Ref.		Ref.		Ref.		Ref.	
Primary	2.30 (1.52–3.47)	<0.001	1.76 (1.17–2.63)	0.006	2.33 (1.52–3.57)	<0.001	1.22 (0.79–1.88)	0.378
Secondary	3.41 (2.26–5.14)	<0.001	2.01 (1.34–3.02)	0.001	4.57 (3.03–6.89)	<0.001	1.47 (0.95–2.27)	0.084
Higher	3.69 (2.44–5.57)	<0.001	2.04 (1.35–3.08)	0.001	5.45 (3.61–8.24)	<0.001	1.41 (0.90–2.22)	0.132
Wealth Index:								
Poorest	Ref.		Ref.		Ref.		Ref.	
Poorer	1.31 (1.19–1.45)	<0.001	1.16 (1.04–1.31)	0.009	1.96 (1.61–2.38)	<0.001	1.57 (1.27–1.94)	<0.001
Middle	1.28 (1.14–1.43)	<0.001	1.08 (0.94–1.25)	0.293	2.52 (2.07–3.07)	<0.001	1.99 (1.55–2.55)	<0.001
Richer	1.45 (1.30–1.61)	<0.001	1.23 (1.06–1.42)	0.007	3.51 (2.82–4.26)	<0.001	2.82 (2.17–3.66)	<0.001
Richest	1.51 (1.34–1.71)	<0.001	1.30 (1.11–1.53)	0.001	3.47 (2.84–4.22)	<0.001	2.74 (2.08–3.61)	<0.001
Health insurance:								
Yes	Ref.		Ref.		Ref.		Ref.	
No	1.19 (1.10–1.29)	<0.001	1.12 (1.04–1.21)	0.002	1.40 (1.22–1.60)	<0.001	1.14 (1.00–1.30)	0.047
Smoked cigarettes in the last 30 days:								
No	Ref.		Ref.		Ref.		Ref.	
Yes	2.13 (1.98–2.28)	<0.001	1.96 (1.83–2.10)	<0.001	3.88 (3.33–4.52)	<0.001	2.87 (2.46–3.35)	<0.001
Fruits and vegetables 5 or more:								
No	Ref.		No include		Ref.		No include	
Yes	1.00 (0.88–1.13)	0.969			1.01 (0.85–1.22)	0.874		
PHQ-9 5 or more:								
No	Ref.		No include		Ref.		No include	
Yes	0.95 (0.85–1.06)	0.336			1.03 (0.89–1.19)	0.686		
BMI:								
Thinness to Normal	Ref.		Ref.		Ref.		Ref.	
Overweight	1.29 (1.18–1.41)	<0.001	1.12 (1.03–1.22)	0.006	1.41 (1.21–1.64)	<0.001	1.30 (1.12–1.50)	<0.001
Obesity	1.51 (1.38–1.66)	<0.001	1.22 (1.11–1.34)	<0.001	1.63 (1.40–1.90)	<0.001	1.50 (1.28–1.76)	<0.001
Hypertension: **								
No	Ref.		No include		Ref.		Ref.	
Yes	0.95 (0.87–1.04)	0.238			0.58 (0.48–0.71)	<0.001	0.91 (0.75–1.11)	0.349
Diabetes history:								
No	Ref.		No include		Ref.		No include	
Yes	0.80 (0.62–1.03)	0.082			0.82 (0.55–1.21)	0.321		

RP: prevalence ratio; aPR: adjusted prevalence ratio; 95% CI: 95% confidence interval. * Estimates included the expansion factor and ENDES sample specifications. ** If the average systolic blood pressure (two readings) was ≥140 mmHg or diastolic blood pressure was ≥90 mmHg or if there was a previous diagnosis by a doctor.

## Data Availability

Publicly available datasets were analyzed in this study. These data can be found here: http://iinei.inei.gob.pe/microdatos/, accessed on 3 November 2021.

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
