# Peer review of "Gender Differences in the Factors Associated with Alcohol Binge Drinking: A Population-Based Analysis in a Latin American Country"

_ijerph, 2022, doi:10.3390/ijerph19094931_

Round 1

Reviewer 1 Report

Dear Authors,

This study is very interesting and important for minimalize of problems and disorders conected with alcohol drinking. 

Authors carefully describe association binge drinking with age, sex and diffrent social-demographic variables in Latin population.

The methodology is transparent and cleary - it is advantage of the manuscript.

Additionally, authors showed main of study limitation for supplement analyses in the future (in the another a researches).

But I have doubt about consent of National Institute of Statistics and Informatics of Peru  to using them data.

Do authors have National Institute of Statistics and Informatics of Peru consent for analyses of data?

The authors should describe of National Institute of Statistics and Informatics of Peru consent for analyses of data in this study.  For example - in methodology, in the part "Ethical considerations" the authors should note: number and date of consent. 

Author Response

Dear Editor and Reviewers,

Thank you for your valuable suggestions and remarks. A. Please find below the point-by-point responses to your comments. Corresponding edits in the main text have been highlighted for your convenience. We hope that the present version of the manuscript will be acceptable for publication, and we look forward to your feedback.

Reviewer #1:

  • This study is very interesting and important for minimalize of problems and disorders connected with alcohol drinking. Authors carefully describe association binge drinking with age, sex and diffrent social-demographic variables in Latin population. The methodology is transparent and cleary - it is advantage of the manuscript. Additionally, authors showed main of study limitation for supplement analyses in the future (in the another a researches). But I have doubt about consent of National Institute of Statistics and Informatics of Peru to using them data. Do authors have National Institute of Statistics and Informatics of Peru consent for analyses of data? The authors should describe of National Institute of Statistics and Informatics of Peru consent for analyses of data in this study.  For example - in methodology, in the part "Ethical considerations" the authors should note: number and date of consent.

Response: Thank you for the recommendation. In the Ethical considerations section, it is detailed that the National Institute of Statistics and Informatics (INEI, for its acronym in Spanish) provides the databases of the Peruvian Demographic and Family Health Survey (ENDES, for its acronym in English) to the public domain and can be downloaded from the website of the INEI (http://iinei.inei.gob.pe/microdatos/). For this reason, INEI's consent was not required to conduct this study.

Reviewer 2 Report

This is a well written paper. The analyses and the writing are clear.

Some suggestions to strengthen the paper.

1) Remove the footnote from Table 1 on binge drinking because binge drinking is not in the table

2) The paper starts, “Alcohol consumption is a public health problem in Peru, fostered by traditional practices.” Perhaps explain what is meant by “traditional practices.”

3) Consider explaining the groups “native,” “non-native,” and “Afro-Peruvian.” Are the “native” indigenous to Peru and do they belong to different groups? Are the non-native mainly descendants of European colonists? Are the Afro-Peruvians the descendants of slaves? In which group are the descendants of Japanese? Who has the lowest social and economic status, and power? What is the population distribution? This would be helpful to an international audience.

4) I don’t really understand the point about vulnerability in lines 92 and 93 of the discussion. It reads to me that the paper is saying that in the United States African American women are the most vulnerable and in Brazil white women are the most vulnerable because both groups binge drink more.

5) It would strengthen the paper to describe a little more the people without health insurance. Higher income and not having health insurance are both positively associated with binge drinking. One would think that those without health insurance have lower income and binge less than those with health insurance. Readers outside of Peru may need more contextual information to understand the findings around health insurance.

Author Response

Dear Editor and Reviewers,

Thank you for your valuable suggestions and remarks. A. Please find below the point-by-point responses to your comments. Corresponding edits in the main text have been highlighted for your convenience. We hope that the present version of the manuscript will be acceptable for publication, and we look forward to your feedback.

Reviewer #2:

This is a well written paper. The analyses and the writing are clear.

Some suggestions to strengthen the paper.

  • Remove the footnote from Table 1 on binge drinking because binge drinking is not in the table.

Response: Thank you for the recommendation. In accordance with the request, the term binge drinking has been removed.

  • The paper starts, “Alcohol consumption is a public health problem in Peru, fostered by traditional practices.” Perhaps explain what is meant by “traditional practices.”

Response: Thank you for the recommendation. In Peru, alcohol consumption is related to traditional or cultural practices that are observed in various regions of the country and are rooted in people. The study carried out by Yamaguchi et al.1, in the Andean region of Peru, reports that people have alcohol consumption that comes from cultural practices and occurs more frequently at community parties and social gatherings, which has normalized excessive consumption of alcohol, especially on weekends. These cultural activities improve social interaction by eliminating stereotypes such as shame, fear and fear. In addition, the consumption of alcohol in the Andean region provides energy and a feeling of warmth, which facilitates field work and the achievement of objectives in certain labor negotiations. On the other hand, offering alcohol to a person could strengthen a bond between people because it is considered a symbol of sincerity, trust and commitment. Also in Peru, excessive alcohol consumption is associated with a lack of control or supervision in different strata of society, where the lack of control of parents with their adolescent children stands out, the culture of "machismo", where men have control over the decisions that are made at home and, consequently, the economic decisions that are based on the purchase and consumption of alcoholic beverages, and the lack of control by the authorities to reduce excessive consumption despite the cultural practices of the people.

To clarify, now in the abstract it states: “Alcohol consumption is a public health problem in Peru, fostered by traditional cultural practices, where promoting social interaction in celebrations, facilitating fieldwork as a source of energy and warmth, and achieving objectives in certain labor negotiations, play an important role.”.

  1. Yamaguchi, S., Lencucha, R. & Brown, TG Control, poder y responsabilidad: un estudio cualitativo de las perspectivas locales sobre los problemas con el consumo de alcohol en las tierras altas de los Andes peruanos. Salud Global 17, 109 (2021).
  • Consider explaining the groups “native,” “non-native,” and “Afro-Peruvian.” Are the “native” indigenous to Peru and do they belong to different groups? Are the non-native mainly descendants of European colonists? Are the Afro-Peruvians the descendants of slaves? In which group are the descendants of Japanese? Who has the lowest social and economic status, and power? What is the population distribution? This would be helpful to an international audience.

Response: Thank you for the recommendation. The ethnicity variable was constructed from the variable "QS25BB" of the ENDES survey, which refers to the ethnic group with which the respondent feels identified and responds to their customs. Within the categories of this variable, there are the autochthonous ethnic groups of Peru, which were categorized as native and include the Quechua, Aimara, and native or indigenous ethnic groups of the Amazon. The non-native category is made up of people who consider themselves white, mestizo or other, which could include people who are descendants of other regions of the world such as Japanese. Lastly, the Afro-Peruvian category refers to people self-identified as black, brown, zambo, mulatto or Afro-Peruavian people, being descendents of people who migrated from Africa to the city of Lima as slaves in the fifteenth century, and who were distributed in various regions of the country 1.

  1. Mori Vilca N. Los afrodescendientes del Perú : historia, aportes y participación en el desarrollo del país. MINEDU, 2018.

To provide greater detail of the aforementioned variable, it now reads: "In the ethnicity variable, the native category is made up of those people who identify themselves as belonging to an indigenous ethnic group in Peru, the non-native category is made up of people who identify themselves as white, mestizo or other groups that are not recognized as natives, while in the Afro-Peruvian category there are people who self-identify as black, brown, zambo, mulatto or Afro-Peruvian people.”

  • I don’t really understand the point about vulnerability in lines 92 and 93 of the discussion. It reads to me that the paper is saying that in the United States African American women are the most vulnerable and in Brazil white women are the most vulnerable because both groups binge drink more.

Response: Thank you for the recommendation. To improve the redaction, now it states “The literature in this regard is variable. Studies in the US population find that non-Hispanic African Americans (55) and specifically African American women (56) had one of the lowest proportions of binge drinking. On the other hand, in Brazil, there has been reported no difference in harmful alcohol consumption patterns in women according to ethnic identification (21). Since ethnicity could be a contributing factor in alcohol binge drinking patterns depending on the type of society, it would be necessary to expand the research of the factors associated with binge drinking in this population, including the influence of this factor”.

  • It would strengthen the paper to describe a little more the people without health insurance. Higher income and not having health insurance are both positively associated with binge drinking. One would think that those without health insurance have lower income and binge less than those with health insurance. Readers outside of Peru may need more contextual information to understand the findings around health insurance.

Response: Thank you for the recommendation. We added the following information in Methods “Regarding health insurance, it was considered yes if the individual was affiliated with Seguro Integral de Salud (for the poor and extremely poor, approximately 45% of the population), or Social Health Insurance (for dependent workers and their legal beneficiaries, approx. 25% of the insured population), or Police, Armed forces or private health insurance providers (approx. 5% of the insured population). In other cases, it was considered no health insurance”. Providing this information, now the reader could understand the implications of the findings: The uninsured population found to have a higher prevalence of binge drinking compared to the insured population are young adults. This finding is confirmed by what was reported in our study on the age of the participants, where the young adult population presents a higher probability of binge drinking as opposed to older adults. These implications were indicated by the authors in the discussion: “In Peru, approximately three-quarters of the population has some type of health insurance and the insured population, are mostly young adults [35]. This finding is confirmed by what was reported in our study on the age of the participants, where the young adult population presents a higher probability of binge drinking as opposed to older adults. This indicates that a subgroup of the population presents a barrier to access health care and has limited access to health promotion and addiction prevention activities necessary in people suffering from binge drinking.”

Round 2

Reviewer 1 Report

Dear Authors,

The answer is clear for me. I understand that everyone can download the data and process them. The key sentence is: "consent is not required to data processing, which was downloaded".

Best regards

Reviewer